# Characterization of Monochromate and Hemichromate AFm Phases and Chromate-Containing Ettringite by $^{1}$H, $^{27}$Al, and $^{53}$Cr MAS NMR Spectroscopy

## Shuai Nie and Jørgen Skibsted *

Department of Chemistry and Interdisciplinary Nanoscience Center (iNANO), Aarhus University, Langelandsgade 140, DK-8000C Aarhus, Denmark; shuainie@chem.au.dk
* Correspondence: jskib@chem.au.dk

**Abstract:** The calcium aluminate hydrate AFm and AFt phases formed upon hydration of Portland cement have an important role in the stabilization and solidification of hazardous chromate ions in hardened cement. AFm monochromate ($Ca_4[Al(OH)_6]_2(CrO_4)\cdot12H_2O$), AFm hemichromate ($Ca_4[Al(OH)_6]_2(CrO_4)_{0.5}(OH)\cdot12H_2O$) and the chromate-containing AFt phase, $Ca_6[Al(OH)_6]_2(CrO_4)_3\cdot24H_2O$, were synthesized and investigated by $^{1}$H, $^{27}$Al, and $^{53}$Cr MAS NMR spectroscopy. $^{27}$Al quadrupolar coupling parameters ($C_Q$, $\eta_Q$) and isotropic chemical shifts ($\delta_{iso}$) were determined for the three phases, including two distinct Al sites in chromate-AFt, as observed by $^{27}$Al MAS and MQMAS NMR. Two dominant peaks are apparent in the $^{1}$H MAS NMR spectra of each of the phases. For the AFm phases, these resonances are assigned to framework hydroxyl groups (1.7–2.0 ppm) and water molecules/hydroxyls (5.0–5.5 ppm) in the interlayer. For chromate-AFt, the peaks are ascribed to framework hydroxyl groups in the $[Ca_6Al_2(OH)_{12}]^{6+}$ columns (~1.4 ppm) and water molecules (~4.8 ppm) associated with the Ca ions. $^{53}$Cr MAS NMR spectra acquired at 22.3 T for the samples show a narrow resonance for both chromate AFm phases, whereas indications of three distinct Cr resonances are apparent for the chromate AFt. The absence of any second-order quadrupolar effects in the $^{53}$Cr NMR spectra strongly suggests that the chromate ions are highly mobile in the anionic sites of the AFm and AFt structures. The NMR data reported in this work are in agreement with the reported crystal structures for the chromate AFm and AFt phases and may be useful for identification and characterization of chromate fixation in cementitious systems, complementing information gained from conventional powder X-ray diffraction studies.

**Keywords:** calcium aluminate hydrates; chromate ions; $^{1}$H NMR; $^{27}$Al NMR; $^{53}$Cr NMR

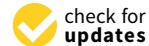



## 1. Introduction

The stabilization/solidification of hazardous waste products by cementitious materials is a widely used approach as a result of the low cost of cement binders, their general availability, and the formation of stable products upon hydration [1–4]. Chromium plays an important role in several industrial products and processes, such as pigments, heterogeneous catalysts, electroplating, and petroleum refining, since it is a redox-sensitive transition element. Chromium can be found in natural soils, refractories, and in many types of waste materials (e.g., fly ashes and metal-ore processing waste). Moreover, it is also present in trace amounts in Portland cement, originating either from the raw materials or alternative fuels (e.g., used oil, tires, plastics, and wastewater sludge) of cement production, where it mainly occurs in the oxidation state Cr(III) [5]. Hexavalent chromium is a highly toxic, carcinogenic, teratogenic, and mutagenic element, reflecting its high solubility and oxidizing potential. In biological systems, chromium can be lethal at levels close to 0.1 mg/g of the body weight [6,7]. A stabilization/solidification of chromium can be achieved in solid matrices, such as the hydration products from ordinary Portland

cement [8–10]; composite cements, including supplementary cementitious materials [11]; or alkali-activated cementitious systems [4,12].

The calcium aluminate hydrates AFm (alumina-ferric oxide-mono) and AFt (alumina-ferric oxide-tri) phases are the two main aluminium-containing hydration products resulting from Portland cement hydration, and they exhibit important roles in the stabilization/solidification of hazardous waste in cementitious systems [2,12,13]. Generally, the AFm phases (Figure 1) have a layered structure with positively charged principal layers composed of $[Ca_4Al_2(OH)_{12}]^{2+}$ and with different types of charge-balancing anions (e.g., $Cl^-$, $OH^-$, $SO_4^{2-}$, $CO_3^{2-}$, and $CrO_4^{2-}$) located in the interlayers. On the other hand, the AFt phases are characterized by two distinct structural components in the form of columns and channels in the intercolumn space. The columns have the composition of $[Ca_6M_2(OH)_{12} \cdot nH_2O]^{6+}$, M = $Al^{3+}$, $Cr^{3+}$, $Si^{4+}$, $Ge^{4+}$, and the channels are occupied by oxyanions ($SO_4^{2-}$, $CO_3^{2-}$, $CrO_4^{2-}$) or water molecules in four crystallographically distinct T sites [14,15].

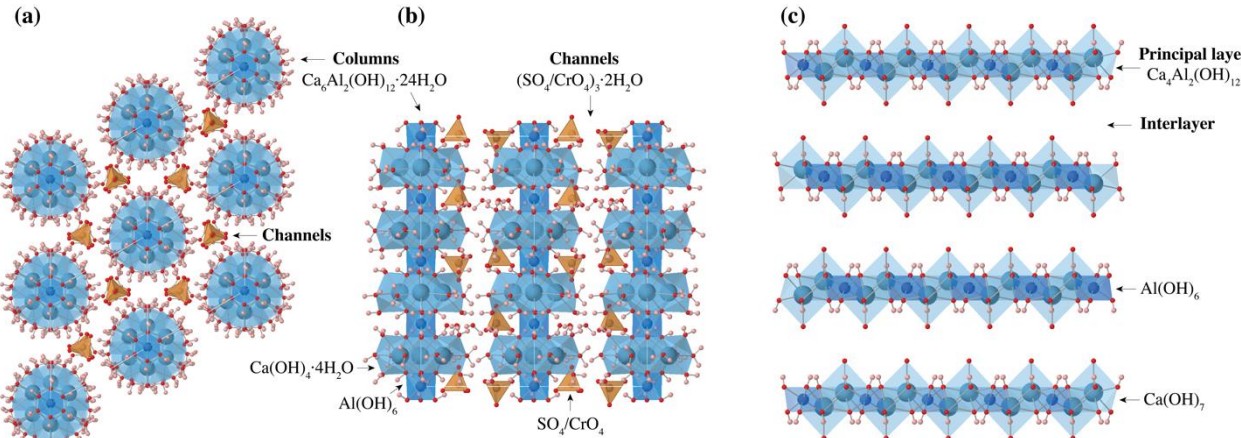

**Figure 1.** Illustrations of the crystal structures for (**a**) AFt (projection along [001]), (**b**) AFt (projection along [100]), and (**c**) AFm (projection along [100]) phases.

The incorporation of chromium in AFm and AFt phases has been investigated in several studies [16–24], mainly using X-ray diffraction (XRD), Raman, and Fourier-transform infrared spectroscopy (FT-IR) as analytical techniques. This includes investigations on the effective removal and fixation of $Cr^{6+}$ ions from solution by AFt [13] and AFm phases, such as Friedel's salt ($Cl^-$ form) [25,26] and monocarbonate ($CO_3^{2-}$ analogue) [21]. The solubilities of chromate AFm ($[Ca_4Al_2(OH)_{12}]^{2+} \cdot [CrO_4 \cdot nH_2O]^{2-}$) and AFt ($[Ca_6Al_2(OH)_{12} \cdot 24H_2O]^{6+} \cdot [(CrO_4)_3 \cdot 2H_2O]^{6-}$) were studied by Perkins and Palmer [17,18], who found that the solubility of chromate-AFt is higher than that of $SO_4$-AFt (ettringite) [27]. This implies that the removal of $Cr^{6+}$ ions from waste water in the presence of sulphate ions by precipitation as ettringite is difficult. The crystal structures of chromate-AFm and chromate-AFt have not been described in detail. Segni et al. [28] reported the structure of chromate-AFm from Rietveld refinement of XRD data using the tetragonal *P*-3 space group. However, a clear location of the $CrO_4^{2-}$ anions was not determined, as the seventh Ca-O bond could either be shared with a water molecule (Ca–O-H$_2$) or the chromate anion (Ca–O-Cr(-O)$_3$), the Ca–O distance being 2.96(3) Å [28]. Recently, Juroszek et al. [23] reported the crystal structure for a new mineral, siwaqaite, ideally $[Ca_6Al_2(OH)_{12} \cdot 24H_2O]^{6+}[(CrO_4)_3 \cdot 2H_2O]^{6-}$, which was found in Jordan using single-crystal synchrotron XRD data. Moreover, Seryotkin et al. [22] proposed the structure for bentorite, $[Ca_6Cr_2(OH)_{12} \cdot 18H_2O]^{6+}[3SO_4 \cdot 2H_2O]^{6-}$. The structures of siwaqaite and bentorite are similar to the crystal structure of ettringite, the main difference being that $Cr^{3+}$ ions replace $Al^{3+}$ in bentorite, whereas $CrO_4^{2-}$ takes the positions of the $SO_4^{2-}$ ions in siwaqaite. For the chromate AFm phases, pure samples of monochromate ($[Ca_4Al_2(OH)_{12}]^{2+} \cdot [CrO_4 \cdot nH_2O]^{2-}$) and hemichromate ($[Ca_4Al_2(OH)_{12}]^{2+} \cdot [1/2CrO_4 \cdot OH-$

$\cdot n\mathrm{H_2O}]^{2-}$) were reported by Pöllmann and Auer [21], including a description of their basic structural features from powder X-ray diffraction.

Solid-state magic-angle spinning (MAS) nuclear magnetic resonance (NMR) spectroscopy is a very powerful technique to investigate the local structure of double-layered hydroxides (LDHs) [29], including the AFm and AFt group phases belonging to the hydrocalumite type of LDHs. For example, the structural α–β phase transition in Friedel's salt (Cl$^-$-AFm) has been characterized by variable-temperature (VT) $^{27}$Al MAS NMR, providing information about the structural changes that occur during the phase transition from an analysis of changes in the $^{27}$Al quadrupolar coupling parameters [30]. For the SO$_4$-AFt phase (ettringite), two distinct Al sites were resolved by $^{27}$Al MAS and multiple-quantum magic-angle spinning (MQMAS) NMR utilizing very high-magnetic-field (22.3 T) instrumentation [31]. This observation supports the trigonal model for ettringite as originally reported from single-crystal X-ray diffraction by Moore and Taylor [32] and most recently refined from single-crystal XRD combined with density functional theory calculations [15,33]. Less detailed solid-state $^{27}$Al NMR studies have also been reported for the hydroxy, sulphate, and carbonate AFm phases [34,35], whereas no investigations have appeared so far for the chromate analogues of the AFm and AFt phases. In this work, monochromate (Cr-AFm), hemichromate (hemi-Cr-AFm), and chromate-AFt (Cr-AFt) were synthesized and characterized for the first time by $^1$H, $^{27}$Al, and $^{53}$Cr NMR spectroscopy. Four different magnetic fields, including a very high magnetic field (22.3 T, 950 MHz for $^1$H), were utilized in the determination of the $^{27}$Al quadrupolar coupling parameters for the chromate AFm and AFt phases, whereas the $^1$H and $^{53}$Cr NMR experiments provide additional information about the hydroxyl groups and water molecules in the structures, as well as the location and dynamics of the $CrO_4{}^{2-}$ ions.

## 2. Materials and Methods

### 2.1. Sample Preparation

The syntheses of chromate AFm (3CaO·Al$_2$O$_3$·CaCrO$_4$·$n$H$_2$O), hemichromate AFm (3CaO·Al$_2$O$_3$·1/2Ca(OH)$_2$·1/2CaCrO4·$n$H$_2$O), and chromate AFt (3CaO·Al$_2$O$_3$·3CaCrO$_4$ ·$n$H$_2$O) were conducted from tricalcium aluminate (3CaO·Al$_2$O$_3$, C$_3$A), CaCrO$_4$ (>99.9%, Alfa Aesar, Karlsruhe, Germany), and CaO in stochiometric proportions. C$_3$A was synthesized from CaCO$_3$ (>99%, Alfa Aesar, Karlsruhe, Germany) and Al$_2$O$_3$ (>98%, Aldrich, Steinheim am Albuch, Germany) using a stochiometric quantity of 3:1 and sintered for 3 h at 1450 °C. CaO was produced by the decarbonation of CaCO$_3$ at 1000 °C for 24 h. The solids were suspended in degassed Milli-Q water with a water/solid ratio of 10 and continuously rotated in sealed high-density polyethylene (HDPE) bottles for 35 days. After reaction, the solids were filtered in a glovebox with an N$_2$ atmosphere to prevent carbonation; subsequently, the residues were dried in a desiccator over silica gel with a slightly reduced pressure for 3 days. The dried samples were stored in sealed glass containers prior to the different types of analysis. The basic structure and purity of the synthesized samples were confirmed by thermogravimetric analysis TGA (Figure A1) and powder XRD analyses (Figure A2).

### 2.2. Characterization Methods

Solid-state $^{27}$Al MAS NMR spectra were acquired on Varian Unity INOVA 300 MHz (7.1 T), Bruker Avance-III HD 400 MHz (9.4 T), Varian Direct-Drive VNMRS-600 MHz (14.1 T) and Bruker Avance 950 MHz (22.3 T) spectrometers using home-built (5 mm at 7.1 T and 4 mm at 14.1 T) or commercial (4 mm at 9.4 T and 2.5 mm at 22.3 T) CP/MAS probes. The single-pulse $^{27}$Al MAS NMR spectra at 7.1 T were recorded with a pulse width of 0.5 μs for an rf field strength of $\gamma B_1/2\pi$ = 65 kHz, using a spinning frequency of $\nu_R$ = 10.0 kHz, a relaxation delay of 2 s, $^1$H decoupling ($\gamma B_2/2\pi$ = 50 kHz) during acquisition, and typically 8192 scans. The $^{27}$Al MAS NMR spectra at 9.4, 14.1 and 22.3 T were obtained in a similar manner, employing $\gamma B_1/2\pi$ = 100 kHz, $\gamma B_2/2\pi$ = 100 kHz, and $\nu_R$ = 10.0 kHz at 9.4 T; $\gamma B_1/2\pi$ = 75 kHz, $\gamma B_2/2\pi$ = 55 kHz, and $\nu_R$ = 13.0 kHz at 14.1 T; and $\gamma B_1/2\pi$ = 100 kHz,

$\gamma B_2/2\pi = 70$ kHz, and $\nu_R = 25.0$ kHz at 22.3 T. $^{27}$Al isotropic chemical shifts were referenced to an external sample of a 1.0 M aqueous $AlCl_3 \cdot 6H_2O$ solution.

The $^{27}$Al MQMAS NMR spectrum (22.3 T, $\nu_R = 25.0$ kHz) of the chromate AFt was obtained with the three-pulse $z$-filter sequence using the $^1$H and $^{27}$Al rf field strengths given above; $^1$H decoupling during the MQ excitation, evolution, and acquisition periods; spectral widths of 45 kHz in both the direct and indirect dimensions; 330 $t_1$ increments and 72 scans for each $t_1$ increment; and a relaxation delay of 2 s. For the $^{27}$Al NMR experiments at 22.3 T, the NMR probe was cooled with air at 5 °C, and relatively low $^1$H rf field strengths were used in order to compensate for frictional heating by the magic-angle spinning and for high-frequency heating by the prolonged $^1$H radiation at 950 MHz. The latter strongly affects the 'zeolitic' water molecules in the AFt structure, as earlier reported for SO$_4$-AFt [31].

The $^1$H MAS NMR experiments (22.3 T) were conducted with a 1.9 mm $^1$H-$^{13}$C-$^{15}$N-$^2$H Bruker MAS probe at MAS frequencies of 35–40 kHz, a 45° excitation pulse for $\gamma B_1/2\pi \approx 60$ kHz, and a relaxation delay of 30 s. $^1$H chemical shifts are relative to neat TMS (Si(CH$_3$)$_4$), using an external sample of adamantane ($\delta_{iso}$ = 1.87 and 1.76 ppm).

The $^{53}$Cr MAS NMR spectra were acquired at 22.3 T using a 4 mm low-gamma X-H probe from Phoenix NMR with a spinning speed of $\nu_R = 12.0$ kHz; a pulse width of 2 µs for an rf field strength of $\gamma B_1/2\pi = 21$ kHz; a relaxation delay of 2; and 40,000 scans for Cr-AFm and Cr-AFt and 120,000 scans for the hemichromate-AFm. A saturated aqueous solution of Cs$_2$CrO$_4$ was used for the pulse-width calibrations and as a secondary reference ($\delta_{iso}$ = 1798 ppm [36]) for the $^{53}$Cr NMR chemical shifts relative to a saturated solution of Cr(CO)$_6$ dissolved in CDCl$_3$. For comparison, a $^{53}$Cr MAS NMR spectrum of solid Cs$_2$CrO$_4$ was also obtained under these conditions.

The powder X-ray diffraction patterns of the synthesized samples were collected on a Rigaku Smartlab diffractometer using an incident monochromator that selects CuKa1 radiation and measuring the 2θ range of 5–70°. Thermogravimetric analyses were conducted on a NETZSCH TG 209 Libra instrument in the temperature range of 50–1000 °C, at a heating rate of 20 °C/min, and with an N$_2$ gas flow of 20 mL/min.

## 3. Results

### 3.1. $^{27}$Al MAS NMR

The $^{27}$Al MAS NMR spectra of the monochromate, hemichromate, and chromate AFt phases recorded at the four different magnetic fields (7.1, 9.4, 14.1, and 22.3 T) are shown in Figure 2 and illustrate the spectral region for the $^{27}$Al central transitions ($m = \frac{1}{2} \leftrightarrow m = -\frac{1}{2}$) of octahedrally coordinated aluminium (AlO$_6$ species). Nearly symmetric resonances are observed for all phases at 9.4 T, with line widths (*FWHM*) of 1.65 ppm (Cr-AFm), 1.94 ppm (hemi-Cr-AFm) and 1.10 ppm (Cr-AFt). At a lower magnetic field (7.1 T), clear indications of a second-order quadrupolar line shape are observed for both the monochromate and hemichromate samples, indicating that these phases possess stronger quadrupolar interactions than chromate-AFt. At higher magnetic fields, the resonances become narrower and shift slightly towards higher frequency, reflecting the inverse proportionality of the second-order quadrupolar broadening and second-order quadrupolar shift with the magnetic field strength and thereby the Larmor frequency ($\nu_L$).

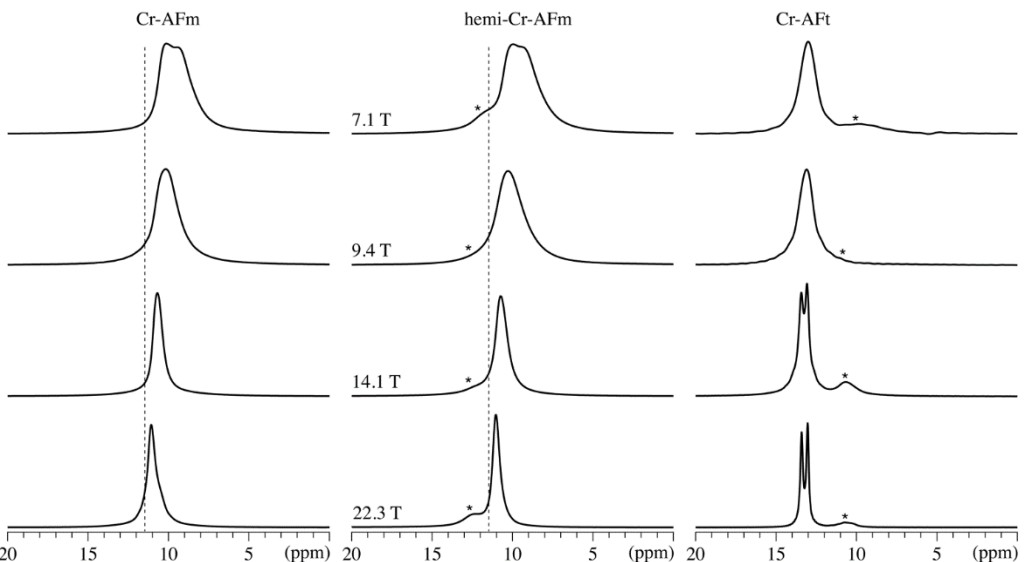

**Figure 2.** $^{27}$Al MAS NMR spectra of the monochromate (Cr-AFm), hemichromate (hemi-Cr-AFm), and chromate AFt (Cr-AFt) samples, illustrating the central transition regions at 7.1 T ($\nu_R$ = 10.0 kHz), 9.4 T ($\nu_R$ = 10.0 kHz), 14.1 T ($\nu_R$ = 13.0 kHz), and 22.3 T ($\nu_R$ = 25.0 kHz). All spectra were acquired with $^1$H decoupling during acquisition. The vertical dash lines show the location of the $^{27}$Al isotropic chemical shifts for Cr-AFm and hemi-Cr-AFm, whereas the asterisks indicate minor resonances from impurity phases (see text).

For the chromate AFt, the resonances at 14.1 and 22.3 T split into two well-resolved peaks, indicating the presence of two distinct $AlO_6$ sites in the crystal structure of Cr-AFt, in agreement with similar observations for the sulphate analogue, ettringite [31]. Minor resonances from impurity phases are also most clearly observed in the spectra at the highest magnetic field. For hemichromate, the resonance at ~12.3 ppm corresponds to a minor impurity of katoite ($Ca_3Al_2(OH)_{12}$), whereas the resonance at 10.5 ppm in the spectra of Cr-AFt is ascribed to a minor amount of monochromate AFm.

Valuable estimates of the $^{27}$Al isotropic chemical shift ($\delta_{iso}$) and second-order quadrupolar effect parameter ($P_Q$) can be determined from the centres of gravity ($\delta_m^{cg}$) for the central transitions and the inner satellite transitions ($m = \pm 1/2 \leftrightarrow m = \pm 3/2$) obtained at the four magnetic fields utilizing the following relationship [37]:

$$\delta_m^{cg} = \delta_{iso} - P_Q{}^2 \frac{C_m}{v_L{}^2}$$
$$C_m = \frac{3}{40} \frac{I(I+1) - 9m(m-1) - 3}{I^2(2I-1)^2} \tag{1}$$

Here, $P_Q = C_Q(1 + \eta_Q^2/3)^{1/2}$, where $C_Q$ is the quadrupole coupling constant, and $\eta_Q$ is the associated asymmetry parameter of the quadrupole coupling tensor (see Table 1 for definitions). The $\delta_m^{cg}$ values ($m = 1/2$ for the central transition and $m = 3/2$ for the inner satellite transitions) at the four magnetic fields are shown as a function of $C_m/v_L^2$ for the chromate-containing phases in Figure 3. The data at 14.1 and 22.3 T are not included for Cr-AFt because the central transition line shapes clearly split into two distinct peaks at these magnetic fields.

**Table 1.** $^{27}$Al isotropic chemical shifts ($\delta_{iso}$) and quadrupolar coupling parameters ($C_Q$, $\eta_Q$) for the chromate AFm and AFt phases determined in this work.

|  |  | $\delta_{iso}$ | $C_Q$ (a) | $\eta_Q$ (a) |
|---|---|---|---|---|
| Cr-AFm |  | $11.26 \pm 0.04$ | $1.10 \pm 0.02$ | $0.16 \pm 0.02$ |
| hemi-Cr-AFm |  | $11.26 \pm 0.04$ | $1.04 \pm 0.02$ | $0.25 \pm 0.02$ |
| Cr-AFt | Al(1) | $13.05 \pm 0.02$ | $0.353 \pm 0.005$ | $0.293 \pm 0.020$ |
|  | Al(2) | $13.44 \pm 0.02$ | $0.359 \pm 0.005$ | $0.245 \pm 0.020$ |
| SO$_4$-Aft (b) | Al(1) | $13.08 \pm 0.05$ | $0.391 \pm 0.010$ | $0.164 \pm 0.020$ |
|  | Al(2) | $13.51 \pm 0.05$ | $0.337 \pm 0.006$ | $0.174 \pm 0.010$ |

(a) The $^{27}$Al quadrupolar coupling parameters are defined as $C_Q = eQV_{zz}/[2I(2I - 1)h]$ and $\eta_Q = (V_{yy} - V_{xx})/V_{zz}$, where $V_{ii}$ represents the principal elements of the electric-field gradient tensor at the nuclear Al site, following the convention: $|V_{zz}| \geqq |V_{xx}| \geqq |V_{yy}|$; $e$ is the charge of the electron; Q is the nuclear quadrupolar moment; $I$ is the nuclear spin-quantum number; and $h$ is the Planck's constant. (b) $^{27}$Al NMR data for SO$_4$-AFt (ettringite) from ref. [31] are included for comparison.

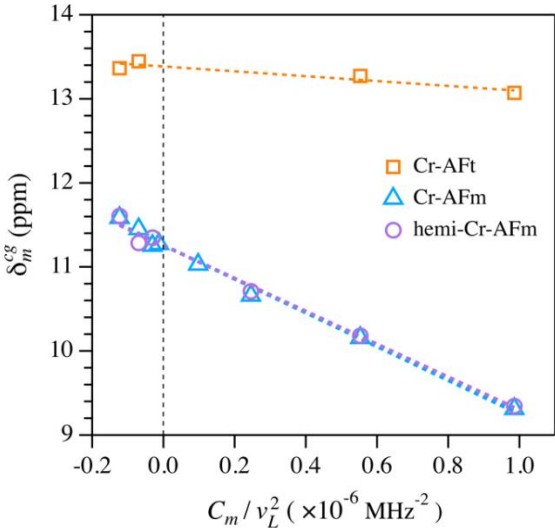

**Figure 3.** Plots of centres of gravity for the $^{27}$Al central transition and inner satellite transitions ($\delta_{1/2,-1/2}^{cg}$, $\delta_{\pm 1/2,\pm 3/2}^{cg}$) for the monochromate, hemichromate, and chromate AFt phases studied at the four magnetic fields (7.1 to 22.3 T), corresponding to the Larmor frequencies $\nu_L$ = 78.03, 104.14, 156.21, and 247.50 MHz.

Linear regression analysis of the data gives the values $\delta_{iso}$ = 11.26 ppm, $P_Q$ = 1.42 MHz for monochromate and $\delta_{iso}$ = 11.26 ppm, $P_Q$ = 1.40 MHz for hemichromate. For Cr-AFt, $\delta_{iso}$ = 13.39 ppm and $P_Q$ = 0.54 MHz is obtained, which should be considered average values for the two Al sites in this phase. The $\delta_{iso}$ and $P_Q$ values for monochromate and hemichromate are nearly identical, and the $P_Q$ values are similar to those reported for other AFm phases (i.e., $P_Q$ = 1.7 $\pm$ 0.2 MHz for monosulphate [34] and $P_Q$ = 1.6 $\pm$ 0.1 MHz ($C_Q$ = 1.42 MHz, $\eta_Q$ = 0.93) for the low-temperature polymorph of Friedel's salt at 18 °C [30]), reflecting that monochromate and hemichromate have similar lamellar structures (Figure 1c) and that the CrO$_4$$^{2-}$ anions only have a moderate impact on the environments of the Al(OH)$_6$$^{3-}$ sites in the principal layers. Furthermore, the strong similarity in $\delta_{iso}$ and $P_Q$ shows that the replacement of half of the CrO$_4$$^{2-}$ anions in monochromate with hydroxyl groups and water molecules in hemichromate only has a very small impact on the local environment of the Al(OH)$_6$ octahedra. This may partly be a result of high degrees of mobility of the species in the interlayer anionic sites, as further supported by the $^{53}$Cr NMR spectra (see below).

The $\delta_{iso}$ and $P_Q$ values represent valuable parameters for a full determination of $\delta_{iso}$, as well as the $C_Q$ and $\eta_Q$ parameters, utilizing the full $^{27}$Al MAS NMR spectrum of the central and satellite transitions as illustrated in Figure 4 for monochromate and hemichromate.

For $^{27}$Al sites experiencing small or intermediate quadrupole couplings, the intensities of the spinning sidebands (ssbs) from the satellite transitions are highly sensitive to the $C_Q$ and $\eta_Q$ parameters [38]. Thus, these parameters can be determined with good precision from least-squares fitting of simulated to experimental ssb intensities for the satellite transitions. These simulations (Figure 4) convincingly reproduce the intensity distributions of the experimental ssb envelopes by using a unique Al site for both monochromate and hemichromate. The optimized $C_Q$ and $\eta_Q$ parameters are summarized in Table 1, along with the $\delta_{iso}$ values obtained from the data in Figure 3. The $C_Q$ values are slightly lower than the estimations from the $P_Q$ values, which may reflect that the measured values for $\delta^{cg}_{m=3/2}$ are influenced by contributions from the outer satellite transitions ($m = \pm 3/2 \leftrightarrow m = \pm 5/2$), resulting in an overestimation of $P_Q$. However, a minute variation in $C_Q$ and a somewhat larger difference in $\eta_Q$ values is observed for monochromate and hemichromate, which reflects that the quadrupole coupling parameters are generally more sensitive to structural changes as compared to $^{27}$Al isotropic chemical shifts.

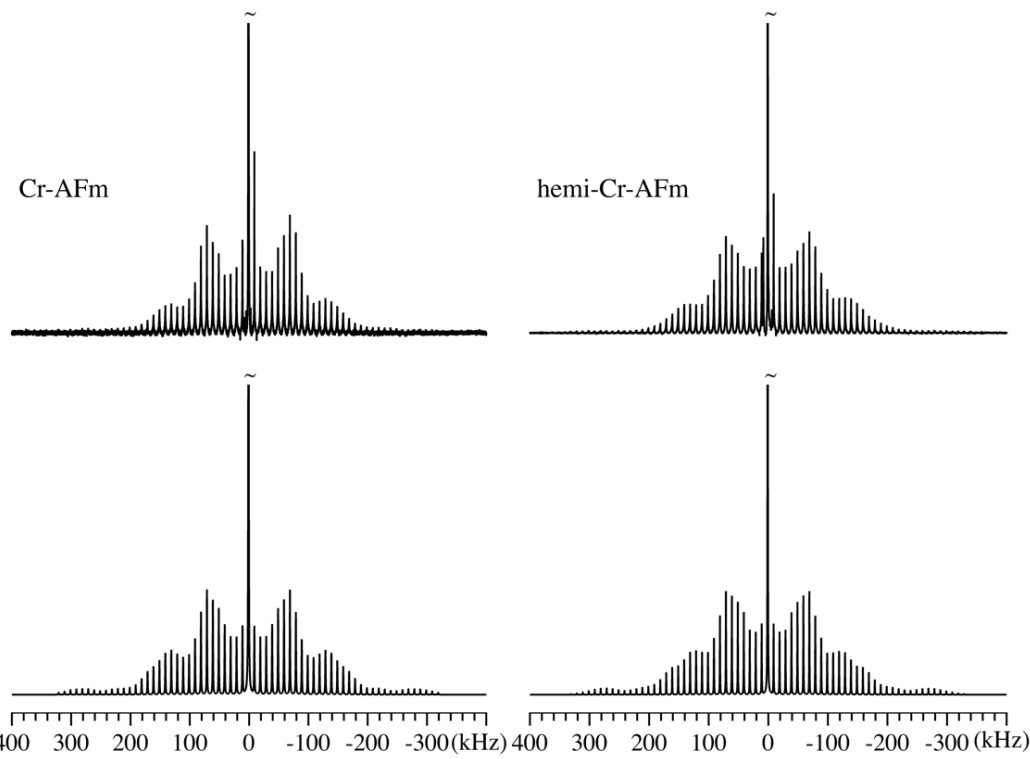

**Figure 4.** Experimental (upper row) and simulated (lower row) $^{27}$Al MAS NMR spectra of the central and satellite transitions for monochromate and hemichromate acquired at 9.4 T using a spinning speed of $\nu_R$ = 10 kHz. The centre band from the central transition is cut-off at approx. 1/20 of its total height in all spectra.

The $^{27}$Al NMR spectra of Cr-AFt obtained at 14.1 and 22.3 T (Figure 2) indicate the presence of two distinct Al sites in the crystal structure, as also found in the triclinic form of ettringite [31]. To further confirm the presence of two such different sites, an $^{27}$Al multiple-quantum magic-angle spinning (MQMAS) NMR spectrum was acquired for this sample at 22.3 T, as shown in Figure 5. This experiment removes second-order quadrupolar broadening in the isotropic (F1) dimension and the contour plot, and both projections clearly reveal resonances from two distinct Al sites in addition to the third peak with low intensity, originating from a minor impurity of monochromate. The dashed line, corresponding to pure chemical shift ($P_Q$ = 0), passes through both peaks, which shows that the two Al sites are mainly distinguished by a difference in $^{27}$Al isotropic chemical shifts.

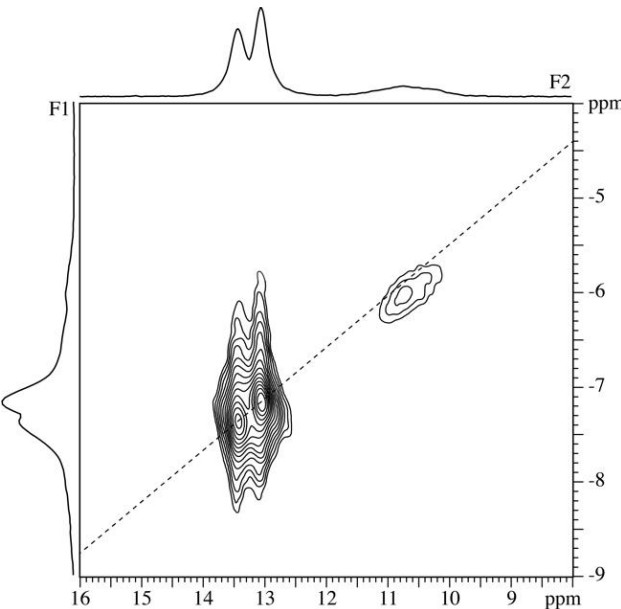

**Figure 5.** $^{27}$Al MQMAS NMR spectrum (22.3 T, $\nu_R$ = 25 kHz) of Cr-AFt acquired with the three-pulse z-filter pulse sequence and $^1$H decoupling during acquisition and the MQ excitation and evolution periods. A spectral width of 45 kHz was used in both dimensions, along with 330 $t_1$ increments in the indirect dimension, 72 scans for each $t_1$ increment, and a relaxation delay of 2 s. The dashed line represents pure chemical shift evolution ($P_Q$ = 0).

A full determination of the $\delta_{iso}$, $C_Q$, and $\eta_Q$ parameters for the two Al sites in Cr-AFt was achieved from analysis of the $^{27}$Al MAS NMR spectrum obtained at 22.3 T (Figure 6), where distinct peaks are also observed for each spinning sideband in the manifold of ssbs from the satellite transitions. Least-squares fitting to the experimental ssb manifold in Figure 6 results in the $^{27}$Al interaction parameters listed in Table 1 for the two Al sites in Cr-AFt. The corresponding optimized simulation (Figure 6) reproduces all details of the experimental spectrum, which warrants a determination of the interaction parameters with high precision. The $^{27}$Al parameters for Cr-AFt are very similar to the values reported for SO$_4$-AFt (Table 1), which suggests that a replacement of SO$_4^{2-}$ by CrO$_4^{2-}$ anions in the channels does not change the main framework structure of the AFt phase. The parameters in Table 1 confirm that the two sites are mainly distinguished by their $\delta_{iso}$ values. This may reflect that the Al sites interact with crystallographically distinct T sites in the channels, as one Al site is surrounded by two SO$_4^{2-}$/CrO$_4^{2-}$ anions, whereas the other site has one SO$_4^{2-}$/CrO$_4^{2-}$ group and water molecules in the nearest vicinity in the channels [15]. However, the local environment and symmetry of the Al(OH)$_6^{3-}$ sites in the main [Ca$_6$Al$_2$(OH)$_{12}$]$^{6+}$ columns (i.e., the electric field gradients at the nuclear Al sites) are less affected by the anions in the channels, as reflected in the very similar $C_Q$ and $\eta_Q$ values for the two Al sites.

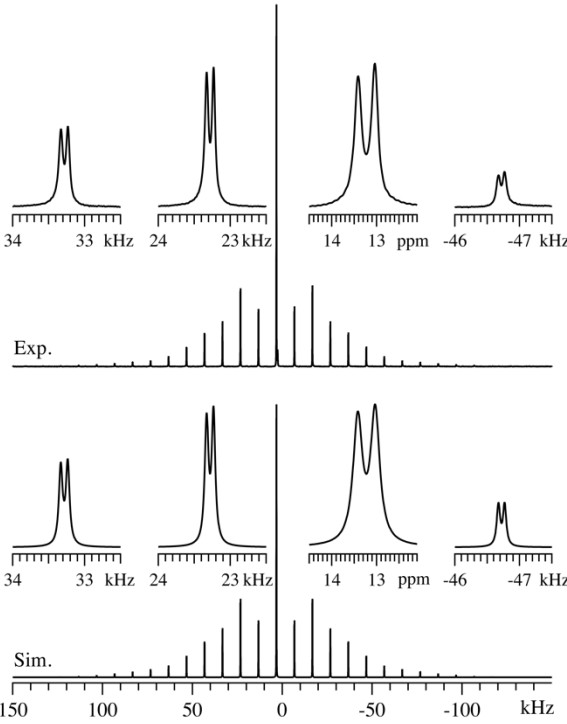

**Figure 6.** Experimental and simulated $^{27}$Al MAS NMR spectrum (22.3 T, $\nu_R$ = 10.0 kHz) of the central and satellite transitions for chromate AFt. The insets show expansions of the central-transition region (ppm scale) and selected spinning sidebands from the satellite transitions (kHz scale), all illustrating the clear resolution of two distinct Al sites.

### 3.2. $^1$H MAS NMR

Despite the natural abundance and high NMR sensitivity of $^1$H, solid-state $^1$H MAS NMR has only been utilized in a few studies of cementitious systems. This mainly reflects the rather small $^1$H chemical-shift range (~15–0 ppm) and the general presence of strong $^1$H–$^1$H dipolar couplings, which results in severe line broadening of the resonances, preventing resolution of different $^1$H structural sites for hydrated systems. However, very fast magic-angle spinning may partly reduce the homonuclear dipolar interactions, and in combination with very high magnetic fields, structural information may be obtained from $^1$H MAS NMR spectra for different types of hydrogen functional groups. For example, this approach has been utilized in studies of the principal layer and interlayers of materials with lamellar structures, e.g., clays [39], layered double hydroxides [40,41], and calcium-silicate-hydrate (C-S-H) phases [42,43].

$^1$H MAS NMR spectra of monochromate, hemichromate, chromate-AFt, and ettringite (SO$_4$-AFt) acquired at 22.3 T with spinning frequencies of 35 or 40 kHz are shown in Figure 7 and represent the first type of such spectra reported for AFm and AFt phases. Generally, these spectra show broadened resonances from two distinct types of local hydrogen environments. For the lamellar chromate AFm phases, the two peaks can be assigned to overlapping resonances from the hydroxyl groups in the principal layers ($[Ca_4Al_2(OH)_{12}]^{2+}$) and hydroxyl groups/water molecules in the interlayer ($[CrO_4 \cdot nH_2O]^{2-}$) by the peaks at approx. 1.8 ppm and 5.0 ppm, respectively. Spectral integration over the two dominant peaks gives intensity ratios of 1:1.15 and 1:1.16 for monochromate and hemichromate, respectively, which are quite close to the expected ratios of 1:1 and 1:1.08 for monochromate ($[Ca_4Al_2(OH)_{12}]^{2+} \cdot [CrO_4 \cdot 6H_2O]^{2-}$) and hemichromate ($[Ca_4Al_2(OH)_{12}]^{2+} \cdot \cdot [(CrO_4)_{0.5} \cdot OH \cdot 6H_2O]^{2-}$), respectively. The hydroxyl groups in the principal layers are bonded to Ca$^{2+}$ and Al$^{3+}$ ions, forming the framework of the lamellar structure of the AFm phases, whereas the water molecules and hydroxyl groups in the interlayer are only weakly bonded to the principal layers. This suggests that the interlayer

water molecules and hydroxyl groups may be mobile and take part in dynamic processes, whereas the framework hydroxyl groups in the principal layer are more rigid. Motional mobility can significantly reduce $^1$H–$^1$H dipolar couplings and thereby result in a reduction in the line widths. Thus, from the line widths of the resonances in the $^1$H NMR spectra of the two AFm phases, the two dominant peaks located at ~1.8 ppm and ~5.0 ppm can be assigned to $^1$H resonances from the principal layer and interlayer, respectively. The line width of the 5.0 ppm resonance from interlayer water molecules/hydroxyl groups for hemichromate is 166 Hz (0.17 ppm), which is much narrower than the corresponding line width for monochromate 422 Hz (0.46 ppm). This indicates that the interlayer water molecules/hydroxyl groups of hemichromate exhibit a higher degree of mobility, which may lead to larger basal spacing of hemichromate as compared to monochromate, as reported earlier for the two chromate AFm phases [21].

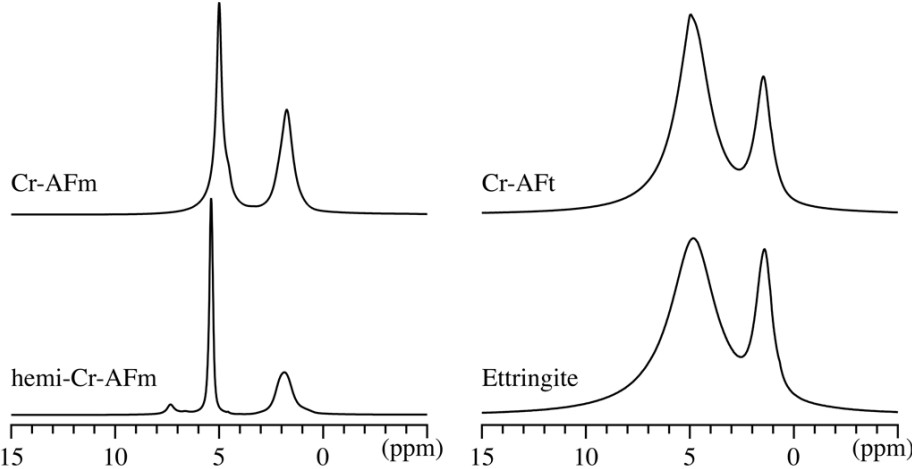

**Figure 7.** $^1$H MAS NMR spectra (22.3 T) for monochromate ($\nu_R$ = 40 kHz), hemichromate ($\nu_R$ = 40 kHz), chromate AFt ($\nu_R$ = 35 kHz), and ettringite ($SO_4$-AFt) ($\nu_R$ = 40 kHz). The main peaks are observed at 5.01 ppm and 1.77 ppm (Cr-AFm); 5.40 ppm and 1.92 ppm (hemi-Cr-AFm); 4.86 ppm and 1.48 ppm (Cr-AFt); and 4.87 ppm and 1.43 ppm (ettringite).

The AFt structure (Figure 1) includes 12 framework hydroxyl groups in the $[Ca_6Al_2(OH)_{12}]^{6+}$ columns, 24 water molecules associated with $Ca^{2+}$ ions, and two zeolitic water molecules in between the columns. The $^1$H NMR spectra of the two AFt phases (Figure 7) show two resonances at approx. 1.4 ppm and 4.9 ppm. Simulation of the partly overlapping peaks gives intensity ratios of 1:3.77 and 1:3.48 for $SO_4$-AFt and Cr-AFt, respectively, which is close to the expected 1:4 ratio for the framework hydroxyls and water molecules associated with Ca. The two highly mobile water molecules between the columns may be absent, as these water molecules are easily removed from the structure at slightly elevated temperatures or low pressure [44]. Thus, following the relative intensities, the two peaks at ~1.4 ppm and ~4.9 ppm are assigned to the hydroxyl groups in the $[Ca_6Al_2(OH)_{12}]^{6+}$ columns and water molecules bonded to Ca, respectively. The line widths of the peak located at ~1.4 ppm for $SO_4$-AFt and Cr-AFt are almost identical, at approx. 800 Hz (0.84 ppm), whereas the line width of the peak from the water molecules is somewhat larger for $SO_4$-AFt, at 2500 Hz (2.63 ppm), as compared to Cr-AFt, at 1834 ppm (1.93 ppm). The much smaller line width of the 1.4 ppm resonance supports the assignment because the hydroxyl groups in the $[Ca_6Al_2(OH)_{12}]^{6+}$ column are more well-ordered compared to the different types of water molecules in the structure. Finally, the very similar line widths observed for $SO_4$-AFt and Cr-AFt suggest that the replacement of $SO_4^{2-}$ by $CrO_4^{2-}$ ions does not lead to significant changes of the hydrogen network in the AFt structure.

### 3.3. $^{53}Cr$ MAS NMR

$^{53}$Cr is a low-gamma quadrupolar nucleus ($I$ = 3/2) with a low natural abundance of 9.5%, a moderate quadrupole moment ($Q$ = −15.0 fm$^2$ [45]), and a Larmor frequency of 53.60 MHz at 22.3 T. This implies that $^{53}$Cr NMR experiments are rather time-consuming, and in some cases, the central transition resonance is so broad that it cannot be observed without distortions in MAS NMR experiments at ordinary spinning frequencies. Thus, a very limited number of solid-state $^{53}$Cr NMR studies have appeared in the literature, mainly focussing on simple chromate and dichromate salts [36,46]. $^{53}$Cr MAS NMR spectra of monochromate, hemichromate, and chromate AFt obtained at 22.3 T are shown in Figure 8. For comparison, a spectrum of Cs$_2$CrO$_4$ is also included, along with a simulation of its quadrupolar line shape, resulting in the parameters $\delta_{iso}$ = 1781 ppm, $C_Q$ = 1.21 MHz, and $\eta_Q$ = 0.22, which are consistent with those reported for Cs$_2$CrO$_4$ by Forgeron and Wasylishen [36]. The $^{53}$Cr NMR spectra for monochromate and hemichromate show only one narrow resonance at 1801.4 ppm (*FWHM* = 154.1 Hz) and 1803.5 ppm (*FWHM* = 185.5 Hz), respectively, with no indications of any quadrupolar broadening. This suggest that both AFm phases include a single Cr site in their crystal structures, with CrO$_4^{2-}$ anions in a highly symmetric environment. Thus, the Cr site is quite different from the chromate environments in Cs$_2$CrO$_4$ and CaCrO$_4$ ($\delta_{iso}$ = 1680 ppm, $C_Q$ = 4.55 MHz, and $\eta_Q$ = 0 [36]), which are both characterized by rather strong quadrupolar interactions. Alternatively, the narrow resonances may occur as a result of dynamic processes that average the quadrupole interaction, suggesting that the CrO$_4^{2-}$ anions are highly mobile in the anionic sites of the AFm structure.

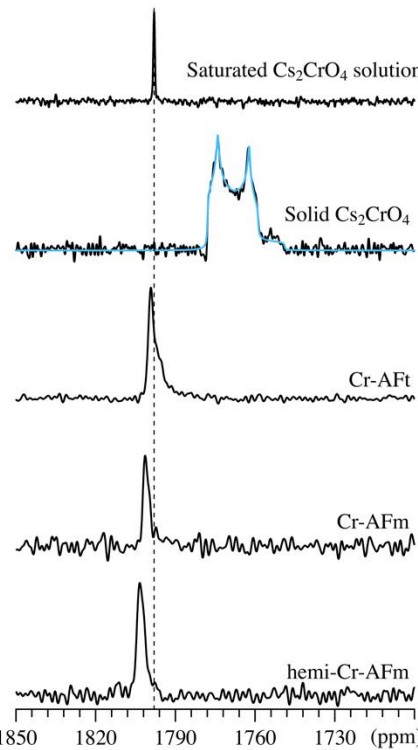

**Figure 8.** $^{53}$Cr MAS NMR spectra (22.3 T) of the central transitions for chromate AFt, monochromate, hemichromate, and a solid sample of Cs$_2$CrO$_4$ acquired with a spinning frequency of $\nu_R$ = 12.0 kHz and a relaxation delay of 2 s. A saturated Cs$_2$CrO$_4$ solution was used as the external chemical-shift reference, which is observed at $\delta_{iso}$ = 1798 ppm relative to the standard reference of a saturated solution of Cr(CO)$_6$ dissolved in CDCl$_3$. The blue line shows the simulated quadrupolar line shape of the $^{53}$Cr resonance for Cs$_2$CrO$_4$, corresponding to the parameters $\delta_{iso}$ = 1781 ppm, $C_Q$ = 1.21 MHz and $\eta_Q$ = 0.22.

A similar feature is observed for the chromate AFt which shows one dominant peak at 1800.7 ppm flanked by two shoulders at 1797.8 and 1795.5 ppm. This indication is in agreement with the crystal structure for $SO_4$-AFt, which includes three distinct S sites [32]. These sites have also be identified by $^{33}$S ($I = 3/2$) NMR, where each resonance showed a significant second-order quadrupolar broadening, corresponding to $^{33}$S quadrupole coupling constants of $C_Q$ = 516, 591 and 810 kHz [47]. The absence of any quadrupolar effects in the $^{53}$Cr NMR spectrum of Cr-AFt suggests that the chromate ions in this phase are highly mobile in contrast to the sulphate groups in $SO_4$-AFt. Increased motional effects in Cr-AFt compared to $SO_4$-AFt may also explain the improved resolution of the two distinct Al sites in the $^{27}$Al NMR spectra (Figures 2 and 6), as mentioned earlier.

## 4. Conclusions

The chromate AFm phases, monochromate and hemichromate, and chromate AFt have been synthesized and characterized by the $^{1}$H, $^{27}$Al, $^{53}$Cr MAS NMR spectroscopy. It is found that monochromate and hemichromate contains a unique Al site in their structures whereas two Al sites are present in Cr-AFt. $^{27}$Al isotropic chemical shifts and quadrupole coupling parameters were determined with high precision for these Al sites, utilizing $^{27}$Al NMR spectra at four different magnetic fields and, in particular, the manifolds of spinning sidebands from the satellite transitions. These parameters are similar to those reported earlier for the AFm phases, monosulphate and Friedel's salt, and the sulphate-AFt phase, ettringite. $^{1}$H MAS NMR spectra of AFm and AFt phases were reported for the first time. For the two AFm phases, it was found that distinct resonances can be observed for the hydroxyl groups in the calcium-aluminate principal layers and the hydroxyls/water molecules in the interlayer. A similar distinction between the hydroxyl groups in the $[Ca_6Al_2(OH)_{12}]^{6+}$ columns and water molecules associated with calcium is observed for $SO_4$-AFt and Cr-AFt. Finally, the $^{53}$Cr MAS NMR spectra of the chromate AFm and AFt phases all show narrow resonances in the range of 1795–1804 ppm, with no indications of any second-order quadrupolar effects for the central transitions. This absence is ascribed to a high degree of structural dynamics for the chromate ions in the AFm and AFt phases. Motional effects are also observed in the $^{1}$H NMR spectra of the AFm phases and in the $^{27}$Al NMR spectra of Cr-AFt—in the latter case, by a higher degree of resolution of the Al sites as compared to $SO_4$-AFt.

**Author Contributions:** S.N.: conceptualization, methodology, software, validation, formal analysis, investigation, writing—original draft, visualization; J.S.: conceptualization, methodology, resources, writing—original draft, supervision, project administration, funding acquisition. All authors have read and agreed to the published version of the manuscript.

**Funding:** The authors acknowledge access to the 950 MHz NMR spectrometer at the Danish Center for Ultrahigh Field NMR Spectroscopy at *i*NANO, Aarhus University, funded by a Ministry of Higher Education and Science grant (AU-2010-612-181). The Carlsberg Foundation is acknowledged for an equipment grant (CF19-0498).

**Data Availability Statement:** The data presented in this study are available within the manuscript.

**Conflicts of Interest:** The authors declare no conflict of interest.

**Appendix A**

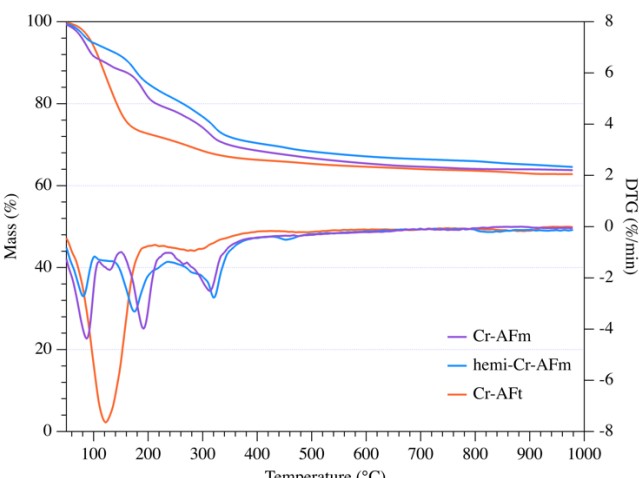

**Figure A1.** Thermogravimetric analysis of the synthesized samples shown as mass-loss and differential curves and obtained on a NETZSCH TG 209 Libra instrument.

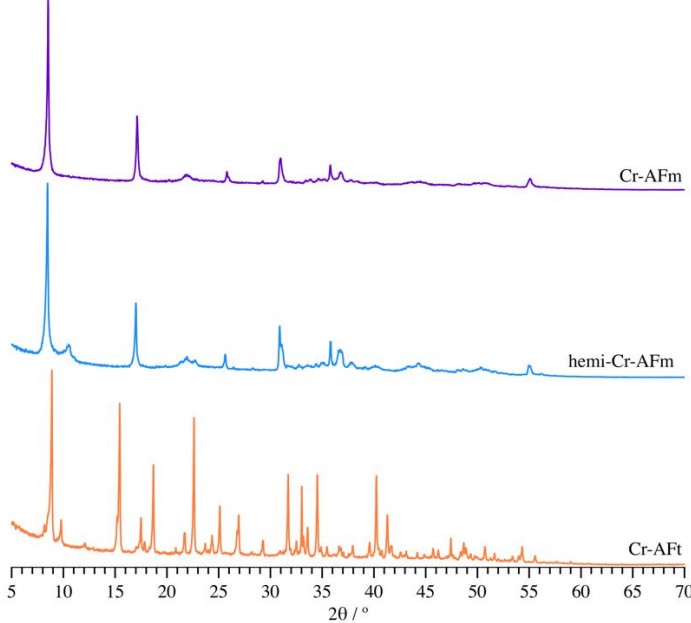

**Figure A2.** Powder X-ray diffraction patterns for the synthesized monochromate and hemichromate AFm phases and chromate AFt collected on a Rigaku Smartlab diffractometer (CuKa1 radiation).

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
