# Peer review of "Characterization of Monochromate and Hemichromate AFm Phases and Chromate-Containing Ettringite by 1H, 27Al, and 53Cr MAS NMR Spectroscopy"

_minerals, doi:10.3390/min12030371_

Round 1

Reviewer 1 Report

The paper reports a characterisation study of Cr-AFm and Cr-AFt using solid state NMR (1H, 27Al and 53Cr). The paper is well written and scientifically well conducted. The paper is undoubtfully worth to be published in Minerals. At the end the structures seem quite similar are those previously determined for CO3-AFm. I just can regret that Rietveld analysis has not been performed on these samples. I just have:

- one request :

You have to add a reference and discuss it (you are forgiven, it is a phD in french) where the structure of Cr-AFm has been determined by Rietveld,  this document is available on line at https://tel.archives-ouvertes.fr/tel-00688377

R. Segni, Caractérisation structurale, propriétés d’échange et stabilité de matériaux de type hydrocalumite [Ca2M (OH) 6]+[Xn 1/n. xH2O]-avec M= Al, Fe, et Sc et X= SO4, CrO4, V2O7 et SiO3, PhD Thesis, Université Blaise Pascal-Clermont-Ferrand II, 2005.

Please check if articles exist, I do not verify this point. The conclusion is quite similar as yours: structure is very close to CO3-AFm.

- and a question about the effect of MAS on water, do you think MAS can affect physically the water in your sample?

Reviewer 2 Report

In this manuscript, the chromate AFm phases, monochromate and hemichromate, and chromate AFt have been synthesized and characterized by the 1H, 27Al, Cr MAS NMR spectroscopy. It is an interesting topic to explore the mechanism of stabilization and solidification of hazardous chromate ions in cement. The paper is well written and clearly presented and it provides some interesting results. There are some issues should be addressed appropriately before it is published. My major comments are as follows:

Comments:

  1. The authors provide regular findings in the abstract, but lack corresponding quantitative conclusions, please supplement in this section.
  2. It is recommended to add more application scenarios in the introduction to clarify the significance of stabilization and solidification of chromium ions in cement.
  3. There are a lot of abbreviations in the manuscript, and some of the abbreviations were not marked with full names, such as “HDPE” in line 106, “TGA” in line 111, and so on. please indicate the full name when the abbreviation first appears, or provide the description of all the abbreviations as attachments.
  4. Please provide more details on sample preparation. In addition, the performance of AFt and AFm is not stable, and it is easy to convert into each other. How could the authors ensure the purity of the sample?
  5. The authors are advised to label the peak positions in Figure 2 for better visualization of chemical shift changes.
  6. The use of “chromate AFt” “Cr-AFt”, “ettringite”, “AFt”, “ettringite (SO42--AFt)” in the manuscript is confusing. Please abbreviate the different types of AFt and express them uniformly.
